# Bicycling Phase Recognition for Lower Limb Amputees Using Support Vector Machine Optimized by Particle Swarm Optimization

**DOI:** 10.3390/s20226533

**Published:** 2020-11-15

**Authors:** Xinxin Li, Zuojun Liu, Xinzhi Gao, Jie Zhang

**Affiliations:** 1School of Artificial Intelligence, Hebei University of Technology, Tianjin 300130, China; lixinxin1128@foxmail.com (X.L.); hebutgxz@163.com (X.G.); 2School of Engineering, Merz Court, Newcastle University, Newcastle upon Tyne NE1 7RU, UK; jie.zhang@newcastle.ac.uk

**Keywords:** lower-limb prosthesis, bicycling, phase recognition, particle swarm optimization (PSO), support vector machine (SVM)

## Abstract

A novel method for recognizing the phases in bicycling of lower limb amputees using support vector machine (SVM) optimized by particle swarm optimization (PSO) is proposed in this paper. The method is essential for enhanced prosthetic knee joint control for lower limb amputees in carrying out bicycling activity. Some wireless wearable accelerometers and a knee joint angle sensor are installed in the prosthesis to obtain data on the knee joint and ankle joint horizontal, vertical acceleration signal and knee joint angle. In order to overcome the problem of high noise content in the collected data, a soft-hard threshold filter was used to remove the noise caused by the vibration. The filtered information is then used to extract the multi-dimensional feature vector for the training of SVM for performing bicycling phase recognition. The SVM is optimized by PSO to enhance its classification accuracy. The recognition accuracy of the PSO-SVM classification model on testing data is 93%, which is much higher than those of BP, SVM and PSO-BP classification models.

## 1. Introduction

As living standards and social welfare improve, lower-limb prosthetics wearers have an increasing demand for various ways of exercise. According to the studies of rehabilitation engineering and the investigation of the lower-limb amputees, bicycle riding is better than walking in terms of physical exercise and avoiding muscle atrophy of the stump. According to the recognized phase, the prosthetic joint could be controlled in an efficient way. In different stages of riding, the above-knee amputee’s joints are in different states, the motor output power is different and the knee joint angle will change. All control strategies are based on the correct recognition of bicycle riding phase. Passive prosthesis [1,2] and semi-active prosthesis [3] cannot actively cooperate with the active movement of the lower-limb amputees. With the development of advanced motor and battery technology, active prosthesis [4] has been developed, making bicycle riding and other forms of exercises possible for lower limb amputees.

In recent years, some advanced prosthetic knee joint products have been developed. The C-LEG prosthetic knee joint, made by Ottobock in Germany, uses angle and torque sensors to judge the motion information of prosthetic wearers. Its self-regulating function can improve the stability of the prosthetic system in the running and bicycle riding of amputees. The 3K180 prosthesis, made by Yoband in United States, can help the prosthetic wearers in bicycle riding and other sports training by its damping-free function. The XC bionic magnetic control knee joint, made by Ossur in Iceland, can enhance the fluency of wears in walking and bicycle riding. Besides, some other prosthetic knee joint products made in Japan and United Kingdom also have good performance [5]. These products have some common features, such as the attached sensors for human motion state detecting, the smart control unit, the high-efficiency motor-driving function, etc.

At present, most scholars in the field of prosthetic control still mainly focus on the recognition and control of walking state [6,7,8,9,10]. Little research in the field of bicycle riding of lower-limb prosthesis wears have been reported. An electric femoral prosthetic controller was developed by Lawson et al. at Vanderbilt University. The recursive least squares method was used in the four-link prosthesis for the prosthetic wearer to achieve the most labor-saving riding [11]. A kind of prosthetic exoskeleton for riding has been designed by Chaichaowarat et al. at Tohoku University in Japan. This prosthesis can store the energy of knee joint contraction in a torsion spring. Then the energy was released when the knee joint is extended. Thus, the fatigue of the leg muscles during repeated contraction exercises is reduced [12]. Sim et al. at Maryland University in USA predicted the maximum speed of bicycle riding by detecting muscle signals, and proposed a neural network-based multiple-input multiple-output controller to improve the coordination of riding [13]. Massoud et al. at the University of Sheffield in United Kingdom proposed PID controllers with different parameters for 6 amputees in the trajectory tracing of bicycle riding [14]. Maaref et al. at University of Alberta in Canada presented a bicycle-type rehabilitation training robot. A method based on Gaussian mixture model is used in this robot. This method is also suitable for the study of prosthetic limb, and has a good reference value [15]. In China, Wang et al. at Shanghai University of Science and Technology realized the distinction between the prosthetic wearer’s riding and walking under different road conditions by combining Gaussian mixture model and hidden Markov model [16]. Liang et al. at Chinese Academy of Sciences used a segmental joint damping control method to regulate the pedaling force of the bicycle-training robot force for people with lower-limb disabilities [17]. However, in these papers, the control of prosthetic knee joint used only one overall algorithm for the whole bicycle period. There is no further analysis on the phase recognition of the bicycle riding process. Actually, the prosthetic knee joint should adopt different control strategies according to the different dynamics features in each bicycle phase. Then, its energy-saving performance and control effect will be improved.

Compared with walking, bicycle riding has different movement feature. In bicycle riding, both feet contact with the pedal, and the alternate movement speed of lower limbs is faster than walking. Moreover, the vibration of bicycle will add the noise into the sensors installed on the prosthesis. In order to improve the accuracy of bicycle riding phase recognition, a multi-sensor system is proposed in this paper. The horizontal and vertical acceleration signals of knee joint and ankle joint, as well as knee joint angle signal, are collected to form multi-dimensional feature vector. Then, these feature vectors are input into a PSO optimized SVM for bicycling phase recognition.

The paper is organized as follows: the bicycle riding process of prosthetic wearers is analyzed in Section 2. Section 3 introduces the data collection and processing process. Section 4 introduces the related recognition methods. Following that, Section 5 presents the results. Finally, conclusions are drawn in Section 6.

## 2. Phase Analysis

Both riding and walking are foot alternating movement. However, there are fundamental differences in the form of movement between the two activities. In bicycle riding, the feet are always in contact with the pedals. While in walking, the foot is suspended in the swaying phase. Therefore, there is no flying state that one foot does not contact the ground when bicycling.

In the recognition of the bicycle riding phase of the lower limb prosthesis, the prosthetic side is considered here. Each phase interval of the prosthetic side in the cycle of riding is discussed. According to the force and energy efficiency during riding, the riding cycle is usually divided into four phases, as shown in Figure 1, where φc is the angle between the pedal crank and the horizontal plane. The energy utilization rate is the highest in the pedaling phase (−45° < φc < 45°). In the relaxation phase (135° < φc < 225°), the pedals will support the prosthesis to move up to the upper buffer phase. In the upper buffer phase (45° < φc < 135°), the pedaling force exert gradually. When reaching the pedaling phase, the exertion reaches maximum.

According to the recognized phase, the prosthetic joint could be control in an efficient way. In the pedaling phase, the amputee would stretch his hip joint of the residual limb and the prosthetic knee joint driving motor should output its highest power for stretch too. In the lower buffer phase, the motor output begins to reduce and then change from stretching to contracting at the maximum knee angle. In the relaxation phase, the joint should be in a damping-free state so that the healthy leg could take the prosthetic leg up in an easy way with less effort. When the upper buffer phase begins, the amputee should draw up his hip joint of the residual limb actively and then the prosthetic knee joint driving motor begins to increase its output at the minimum knee angle. All the control strategies are based on the correct recognition of bicycle riding phases.

However, in the daily bicycle riding process, it is impossible to directly measure φc through the sensor. So, it is necessary to recognize the four phases in the bicycle riding process via other sensor signals installed on the prosthetic wearer.

## 3. Data Acquisition and Preprocessing

### 3.1. Sensor System

In the field of human activity recognition, many kinds sensor systems are proposed [18,19,20,21]. However, some sensors are not suitable for long-term practical application because multiple sensors might cause inconvenience to amputees. Comparatively, some new wireless wearable sensors are now available, which are suitable for long-term activity monitoring and achieve good recognition results for some basic activities, such as lying, walking, and running.

The surface electromyography (s-EMG) signal can reflect the amputee’s intention of motion in advance. However, the s-EMG signal is too weak to collect. Sometimes the surface electrodes are inevitably affected by the sweat. Moreover, the vibration of bicycle and stump limb also leads to poor signal to noise ratio. Furthermore, individual differences of s-EMG signals are obvious for different amputees because of their degree of amputation and muscle atrophy [22]. It shows that the recognition and control based on s-EMG has poor generality for different amputees, even the same one in different physical state. Therefore, in this paper, an inertial measurement unit (IMU) sensor system of kinematics signals is designed to collect the horizontal and vertical acceleration signals at the knee joint and ankle joint of the prosthesis wearer, as well as the knee joint angle signal.

Then, these signals are used for the bicycle riding phase recognition of the prosthetic wearer. In order to reduce the inconvenience of amputees, some light wireless wearable sensors, as shown in Figure 2, are placed on the knee joint and ankle joint. A Delsys sensor is used to collect acceleration signals. A MPU9250 angle sensor is used to collect the angle of the knee joint signal. The calculation of the MPU9250 sensor is based on nine-axis signals. This sensor including three acceleration signals, three angular velocity signals, and three geomagnetic signals. It can calculate the collected knee joint angle data internally. Then, these data are transmitted into the host computer without further calculations.

### 3.2. Data Preprocessing

The 5 above-knee amputee volunteers in the experiment are 165–175 cm high and 25–40 years old, 3 males and 2 females. They all have worn the Ottobock prosthesis for more than 5 years. In order to protect the safety of volunteer, a bicycle was placed on a shelf for data collection. The volunteers are asked to ride at an average speed for five minutes. The average speed is chosen by its own habits. As shown in Figure 3.

Due to the vibration of bicycle and stump limb, noise and interference in the collected signal are inevitable, especially in the critical point from the upper buffer phase to the pedaling phase, and the point from the lower buffer phase to the relaxation phase. As a result, the feature signal might be unclear or even lost. In order to retain the original signal features, the combined soft-hard threshold method, which is proposed by Dnnoho and Johnstone [23], is used to denoise the bicycle riding signal. The improved wavelet packet denoising method is adopted to process the cycling information. This method can be summarized as combining the original single hard threshold or soft threshold denoising method. Firstly, the variance of noise signal and the wavelet transform coefficient are used to determine the threshold value. Secondly, the soft-hard threshold denoising function is used to determine the threshold value of wavelet packet coefficient. Furthermore, the optimal solution of wavelet packet coefficient is obtained. Finally, the denoised signal is constructed by inverse transform with the decomposition coefficient of low-level wavelet packet, in which the noise reduction function is as follows:(1)$$\alpha =\tau \sqrt{2\mathrm{lg}\left(D\right)}$$
(2)$$G_{\left(a,b\right)}=\left\{ \begin{array}{l} \mathrm{sgn}(g_{\left(a,b\right)})(\left|g_{\left(a,b\right)}\right|-k\alpha ),\left|g_{\left(a,b\right)}\right|>\alpha ,0\leq k\leq 1\\ 0,\left|g_{\left(a,b\right)}\right|\leq \alpha ,0\leq k\leq 1 \end{array}\right.$$
where *α* represents the threshold value; *τ* is the variance of the noise signal; *D* is the original signal wavelet packet transform coefficient; *G*_(*a*,*b*)_ is the wavelet coefficient of the original signal after soft-hard threshold processing; *g*_(*a*,*b*)_ is the wavelet coefficients after decomposition of the noisy signal; and *q* is the soft adjustment factor. In Equation (2), *G*_(*a*,*b*)_ obtains its hard threshold when *k* is 0, its soft threshold when *k* is 1, and the combined soft-hard threshold when *k* is between 0 and 1. According to a large number of experiments, setting *k* to 0.5 can effectively reduce the noise level and retain the characteristics of the original signal.

Figure 4, Figure 5, Figure 6 and Figure 7 show the comparison of the original motion signals and the denoised signals by the combination of soft-hard thresholds. In the figure, *t* is the sampling time, in seconds; *Kx* and *Kxl*, respectively, represent the original signal of knee joint horizontal acceleration and the signal after denoising; *Ky* and *Kyl*, respectively, represent the original signal of the vertical acceleration of the knee joint and the signal after denoising, *Ax* and *Axl*, respectively, represent the original signal of the horizontal acceleration of the ankle joint and the signal after denoising, *Ay* and *Ayl*, respectively, represent the original signal of the vertical acceleration of the ankle joint and the signal after denoising.

Figure 8 shows the knee angle signal in bicycle riding. As the MPU9250 has the self-attached interference-proof function, no noise reduction processing is required for the knee joint angle.

### 3.3. Signal Feature Extraction

The joint acceleration is easily affected by external factors in the experiment, and its characteristic value is not easy to be extracted directly. In order to recognize the four bicycle riding phases of the prosthetic wearer effectively, the characteristic value of the knee joint angle at each phase conversion point needs to be determined first. Then the acceleration signal of each joint is analyzed. According to the static experiment, the knee joint angle of the beginning point of the pedaling phase is 84–87°; the knee joint angle of the beginning point of the lower buffer phase is 106–108°; the knee joint angle of the beginning point of the relaxation phase is 95–98°; and the knee joint angle of the beginning point of the upper buffer phase is 73–76°. Compared with different prosthetic wearers, the seat height will be adjusted before riding. After the riding survey of several knee prosthetic wearers participating in the training, it is found that although the final seat height is different due to factors such as height, the angle difference of the knee joint in each phase is very small. Because the same knee joint angle appears twice in the process of leg contraction and stretching in a complete riding cycle, there is no one-to-one mapping relationship between the above angle eigenvalues and the phase transition point. For example, the knee joint angle in the relaxation phase and pedaling phase might be both 96.86°. Therefore, in order to distinguish the riding phase, the acceleration characteristic information of the prosthesis joint is also needed.

In order to improve the accuracy of riding phase recognition and avoid the uncertainty of single acceleration signal, from the perspective of data redundancy, five eigenvalues are selected to form the vector *Pi*(*x*) = [*kx*,*ky*,*hx*,*hy*,*a*] for phase recognition, in which *kx* and *hx* represent the horizontal acceleration of knee joint and ankle joint, *ky* and *hy* represent the vertical acceleration of knee joint and ankle joint, respectively, and a represents the angle of knee joint.

## 4. Riding Phase Recognition Method

### 4.1. Support Vector Machine

Support vector machine (SVM) is a machine learning algorithm originated by Vapnik (standard template library, STL) [24,25,26]. Support vector machine transforms low-dimensional space vector into high-dimensional space. Based on the principle of maximum distance from the hyperplane, it uses the training set to establish the optimal hyperplane in the high-dimensional space. It could make the classification results have strong robustness and high classification accuracy. Therefore, support vector machines are widely used in multi-dimensional data classification. The decision function of SVM is as follows:(3)$$f\left(P\right)=\mathrm{sgn}\left(\sum_{i=1}^{l}Q_{i}\alpha_{i}^{*}K\left(P_{i},P\right)+b^{*}\right)$$
where *P_i_*(*x*) = [*kx*,*ky*,*hx*,*hy*,*a*]; *P*(*x*) is obtained by transforming the input *x* of the feature vector from low-dimensional space to high-dimensional space, that is: *x*→*P*(*x*) = [*P*_1_(*x*),*P*_2_(*x*), …, *P_n_*(*x*)]^T^; *Q*_i_ ∈ {±1}, *i* = 1, 2, 3, …, *n*; *α** is the optimal solution of the training sample and *b** is the threshold.

As the data of bicycle riding are non-linear, it is necessary to transform the data from low-dimensional space to high-dimensional space through kernel function. SVM completes complex operations in low-dimensional space through nonlinear transformation, and establishes an optimal classification surface in high-dimensional space, and this nonlinear change is completed by the kernel function. Therefore, the selection of the SVM kernel function is very important.

The kernel bandwidth *γ* is an important parameter of the kernel function, which affects the complexity and classification accuracy of the SVM model. The penalty factor *C* has the effect of adjusting the complexity of the model and can effectively improve the generalization performance of the model. The changes of the penalty factor *C* and the kernel bandwidth parameters *γ* can affect the classification accuracy of the SVM model. Therefore, different *C* and *γ* are used for comparative testing. The final result shows that the changes in the penalty factor and kernel function bandwidth have a significant impact on the model classification accuracy. The recognition rates for different kernel functions, different kernel bandwidth and penalty factors are shown in Table 1.

After comparing the experimental results, it can be seen that the Gaussian Kernel Function (RBF) has a high recognition rate during the recognition process, which can be maintained above 79%, and is relatively stable. Therefore, the Gaussian kernel function (RBF) is selected in this paper. Then the optimal hyperplane is established to realize the classification of nonlinear data.
(4)$$F=\frac{1}{K}\sum_{l=1}^{K}\frac{P_{lr}}{p_{lr}+p_{lw}}$$
where *P_i_*(*x*) = [*kx*,*ky*,*hx*,*hy*,*a*] is the input of the kernel function.

The kernel function is used to transform the vectors from low-dimensional space to high-dimensional space. In the high-dimensional feature space, binary classification model is used to classify the riding data of prosthetic wearers.

### 4.2. Binary Classification

The multi-dimensional feature vector *P_i_*(*x*) = [*kx*,*ky*,*hx*,*hy*,*a*] is formed by extracting the feature of the riding signals of the prosthetic wearer. Two SVM model are established for multi-classification, as shown in Figure 9.

As shown in Figure 9, plan of binary tree classification model requires classification processing three times. The three SVM classifiers are the same. Its accuracy is high. For plan of secondary classification model, although it only requires classification processing two times, its accuracy is poor. It is easier to find an optimal hyper-plane for only one group of data from the other three groups in the high-dimensional space. Meanwhile, it is more difficult to find the ideal hyper-plane that could divide the four groups of data into two categories. Therefore, the plan of binary tree classification model is used in this paper, as shown in Figure 10.

The phase recognition of bicycle riding is performed as follows. Separate the data in the pedaling phase by using SVM1 first. Then separate the data in the lower buffer phase by using SVM2. Finally, separate the data in the relaxation phase by using SVM3, and the rest one is the data in the upper buffer phase.

### 4.3. Particle Swarm Optimization

In order to improve the efficiency and accuracy of SVM classification model, particle swarm optimization (PSO) [27] is used to optimize the parameters of SVM kernel function. The PSO initializes a group of particles in vector space, where each particle represents a solution to the optimization problem and is represented by position, velocity and fitness. In the solution space, the updating of a single position is completed by tracking individual best value and group best value. Then the fitness value of individual position is updated. Finally, by comparing the best fitness value of new particles and individual particles, as well as the best fitness values of the group, the position of updating individual best values and group best values is determined.
(5)$$V_{p}^{k+1}=\omega V_{p}^{k}+c_{1}r_{1}\left(p_{p}^{k}-x_{p}^{k}\right)+c_{2}r_{2}\left(p_{gd}^{k}-x_{p}^{k}\right)$$
(6)$$x_{p}^{k+1}=x_{p}^{k}+V_{p}^{k}$$
where *r*_1_ and *r*_2_ are independent random numbers between 0 and 1; *c*_1_ and *c*_2_ are acceleration constants representing the weight of local and global optimal position acceleration, respectively; *k* is the number of iterations; *V_i_* is the current particle velocity; *X_i_* is the current particle motion position; *P_i_* is the position of the particle’s local optimal solution; *P_g_* is the position of the particle’s global optimal solution; *ω* is the momentum parameter, the size is related to the objective function and fitness function.
(7)$$\omega =\left\{ \begin{array}{ll} \frac{\left(\omega_{\max}-\omega_{\min}\right)\left(K-K_{\min}\right)}{K_{\mathrm{avg}}-K_{\min}}+\omega_{\min} & K\leq K_{\mathrm{avg}}\\ \omega_{\max} & K>K_{\mathrm{avg}} \end{array}\right.$$
where, *ω*_max_ and *ω*_min_, respectively, represent the maximum inertia coefficient and the minimum inertia coefficient; *K*_min_ represents the minimum value of the fitness function; *K* represents the fitness function; and *K*_avg_ represents the average value of the fitness function.

In this paper, PSO algorithm is used to optimize the parameters of SVM kernel function. Furthermore, the ten-fold cross-validation recognition rate in SVM classification is selected as the fitness function of PSO:(8)$$H=\frac{1}{K}\sum_{l=1}^{K}\frac{P_{lr}}{p_{lr}+p_{lw}}$$
where *K* is the optimized sample number set, which is set as 10 after verification. *P_lr_* represents the samples that were correctly classified in the *l*-th verification; *P_lz_* represents the samples that are misclassified in the *l*-th verification. The larger the *K* value is, the more obvious the optimization effect could be obtained.

As shown in Figure 10, in order to distinguish the four bicycle riding phases, the parameter optimization training is needed for three kernel functions. According to the repeated experiments and comparisons, the population size is set as 20; the maximum number of iterations is set as 200; the local acceleration factor is 1.5; and the global acceleration factor is 1.7. In all the 10 groups data collected in the experiment, 1 group is used as the test set and 9 groups are used as the training set. Ten-fold cross-validation is carried in the data training. The results are shown in Figure 11:

## 5. Experimental Results

In order to verify the validity and reliability of SVM classification model based on PSO optimization for prosthetic wearer phase recognition in bicycle riding, the method is compared with the BP neural network, PSO-BP neural network and SVM.

In the experiment, the data of 50 riding cycles are collected. Each cycle data contains the information of four phases in bicycle riding. Cross-validation was used in this experiment. Therefore, there are 200 groups of riding phase information available. A total of 100 groups of data are selected as training data for the classification model. The remaining 100 groups of data are used as testing data.

BP neural network is trained and tested by data feature sets of known categories [28]. The BP neural network first sends the data to the input layer, and then passes it to the output layer through the hidden layer. Finally, the corresponding value and the global error between the value and the expected value are obtained in the output layer. In the continuous iteration process, the global error value is compared with the expected value. The set expected values are compared, and when the set conditions are met, the iteration is stop, the connection weights and thresholds of each node in the BP neural network are output to establish a neural network classification model. For BP neural network classification model, the parameters are set as: five input layer nodes; six hidden layer nodes; four output layer nodes. Five input layer nodes represent five signal feature values, and four output nodes correspond to four phases. The number of hidden layer nodes is selected according to the Equation (9):(9)$$m=\sqrt{n+l}+\alpha$$
where *n* is the number of input layer nodes; *l* is the number of output layer nodes; *α* is a constant, which is set as 3 after verification.

The learning rate is 0.7; and the hidden layer activation function is selected as sigmoid function. For PSO-BP neural network classification model, the PSO is used to optimize the connection weights and thresholds in the BP neural network to improve the recognition rate of the classification model.

The comparison results on the testing data are shown is Figure 12. In the figure, 1 represents the pedaling phase; 2 represents the lower buffer phase; 3 represents the relaxation phase; 4 represents the upper buffer phase. The circle in the figure indicates the actual phase. Furthermore, the + indicates the recognized phase by each method. If the two symbols are coincident, the recognition is correct. Otherwise, it is wrong. The recognition rate of BP neural network classification model is 72%. That of SVM classification model is 79%. That of PSO-BP classification model is 84%. Furthermore, that of PSO-SVM is 93%. The number of recognitions for each bicycle riding phase is shown in Table 2. The precision of each SVM classification is shown in Table 3. The recall of each SVM classification is shown in Table 4. The F1-score obtained by each SVM classification is shown in Table 5, and the G index is shown in Table 6. The PSO-SVM classification model shows the best correct rate.

Although the parameters of BP neural network can be optimized by PSO algorithm, the structure of BP neural network is based on empirical risk minimization, and its network connection weights are easy to fall into local minimization, which makes the training process appear under learning or over learning state with the change of prediction error. SVM is based on the principle of structural risk minimization. Compared with the traditional BP neural network, SVM has higher generalization performance and stability.

As shown in Table 2, PSO-SVM has a high recognition rate. As shown in Table 3, PSO-SVM has the highest precision and the smallest standard deviation among the four classification models. As shown in Table 4, PSO-SVM has a high recall. As shown in Table 5 and Table 6, the PSO-SVM classification model has a high F1-score and G index. The above results show that the method of PSO optimized SVM is effective in the riding phase recognition. It provides practical information for the control of prosthesis. 

It has important practical significance for the control of lower limb prosthesis in bicycle riding. The prosthetic knee joint can adjust its driving force according to the identified riding phase. This can make the riding action more fluent, higher energy utilization, and save the amputee’s physical strength. The prosthetic knee joint can regulate its driving power according to the recognized riding phase, which makes the riding motion fluently with higher energy efficiency and save the physical strength of amputees at the same time.

## 6. Conclusions

In this paper, a PSO-SVM multi-binary tree classification model is proposed for the recognition of bicycle riding phases. It could provide necessary information for the enhanced control of prosthetic knee joint. The recognition accuracy of the proposed method reaches 93%, which is higher than those from SVM, BP neural network, and PSO-BP classification model. It could make amputees ride bicycle more easily. In future works, the propose method will be integrated with the coordinative control of prosthetic knee-ankle-toe in bicycle riding [29]. Furthermore, the safety protection for the amputees in the bicycle riding will be investigated.

## Figures and Tables

**Figure 1 sensors-20-06533-f001:**
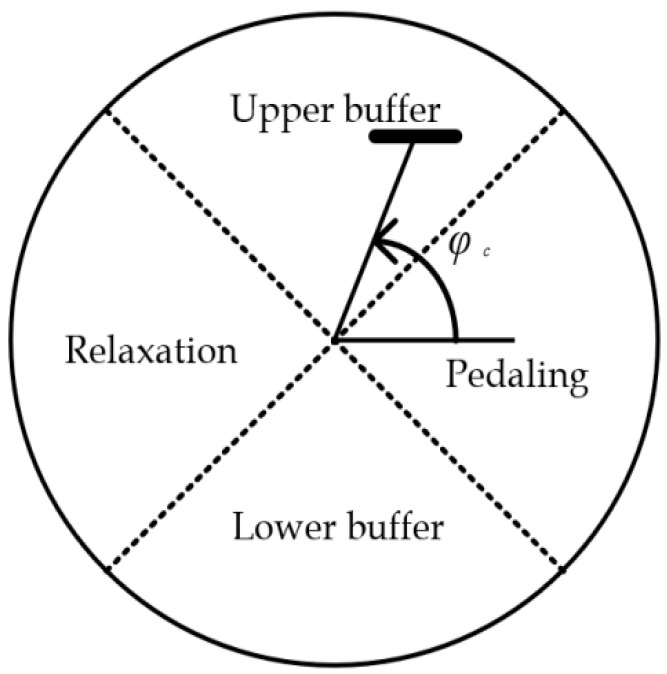
Bicycling phases.

**Figure 2 sensors-20-06533-f002:**
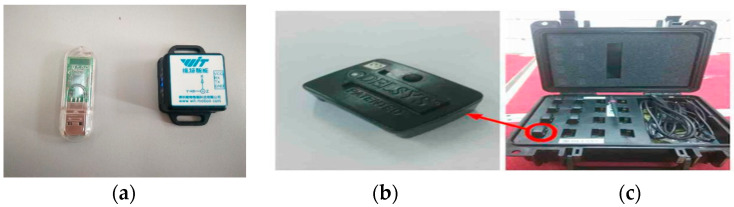
Sensors for motion data collecting. (**a**) MPU9250 angle sensor; (**b**) Delsys accelerator; (**c**) Wireless acquisition system.

**Figure 3 sensors-20-06533-f003:**
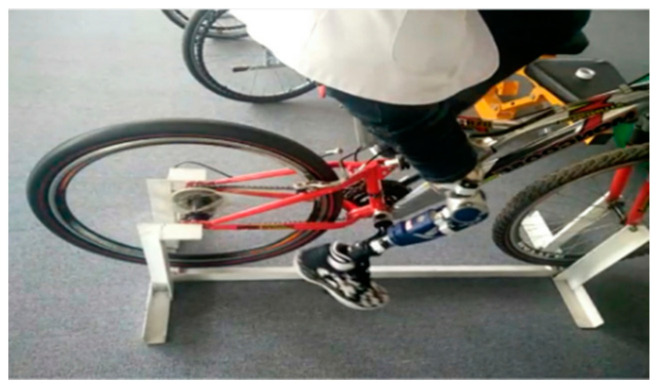
Bicycle riding of prosthetic wearers.

**Figure 4 sensors-20-06533-f004:**
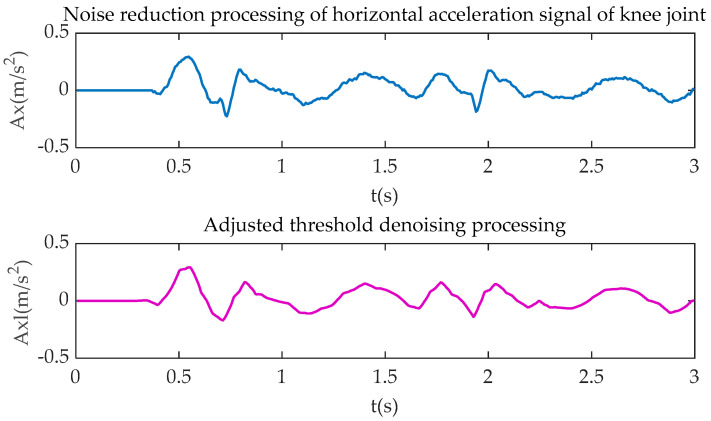
Knee joint horizontal acceleration processing.

**Figure 5 sensors-20-06533-f005:**
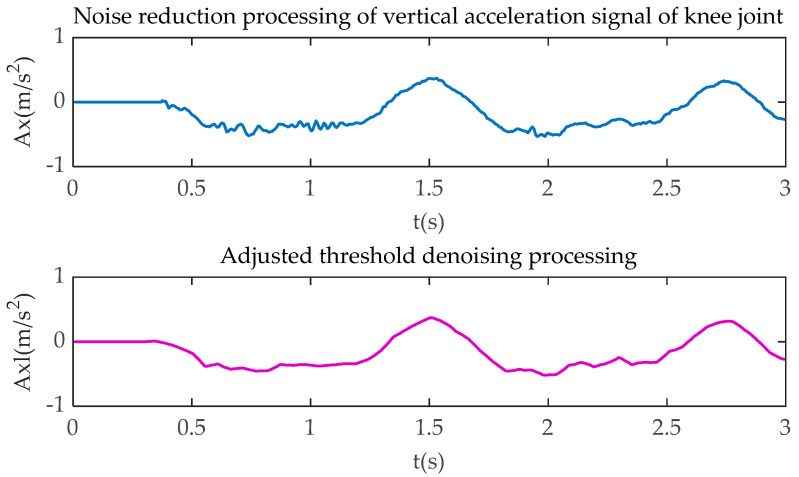
Knee joint vertical acceleration processing.

**Figure 6 sensors-20-06533-f006:**
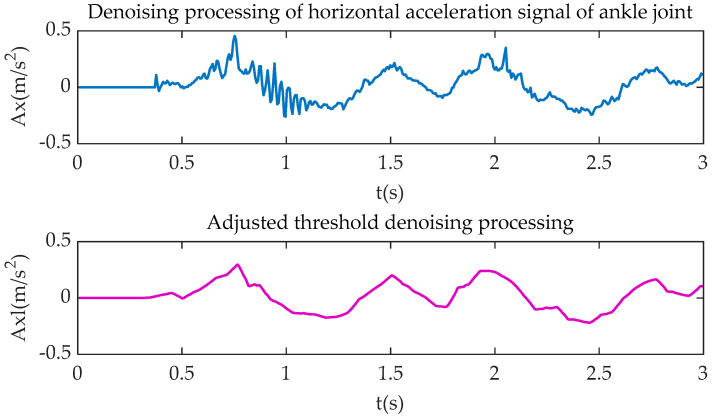
Ankle joint horizontal acceleration processing.

**Figure 7 sensors-20-06533-f007:**
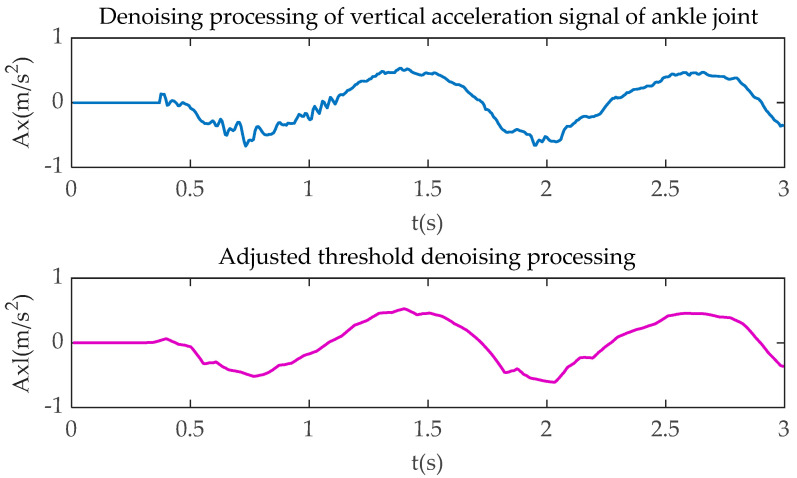
Ankle joint vertical acceleration processing.

**Figure 8 sensors-20-06533-f008:**
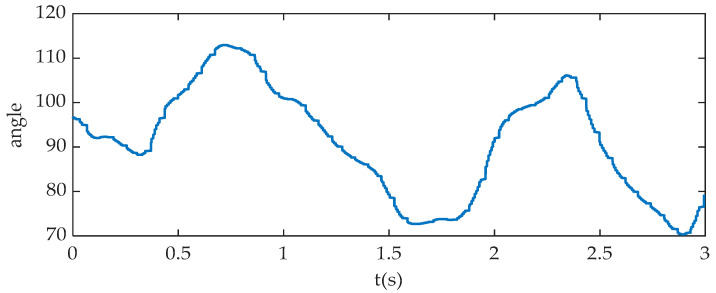
Knee angle signal.

**Figure 9 sensors-20-06533-f009:**
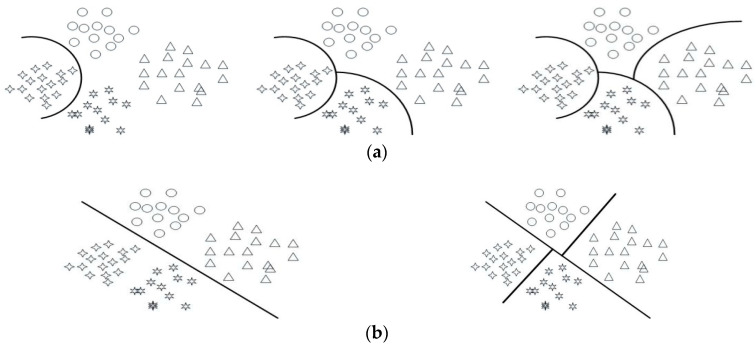
Multi-class model diagram. (**a**) Binary tree classification model; (**b**) Secondary classification model.

**Figure 10 sensors-20-06533-f010:**
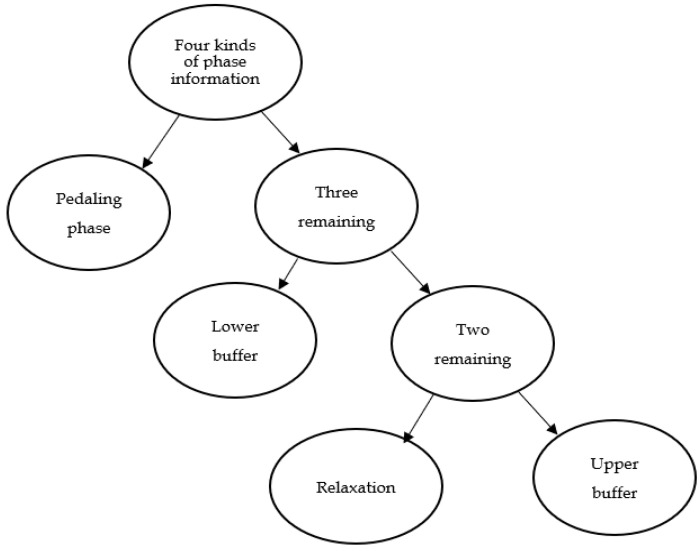
Schematic diagram of binary classification.

**Figure 11 sensors-20-06533-f011:**
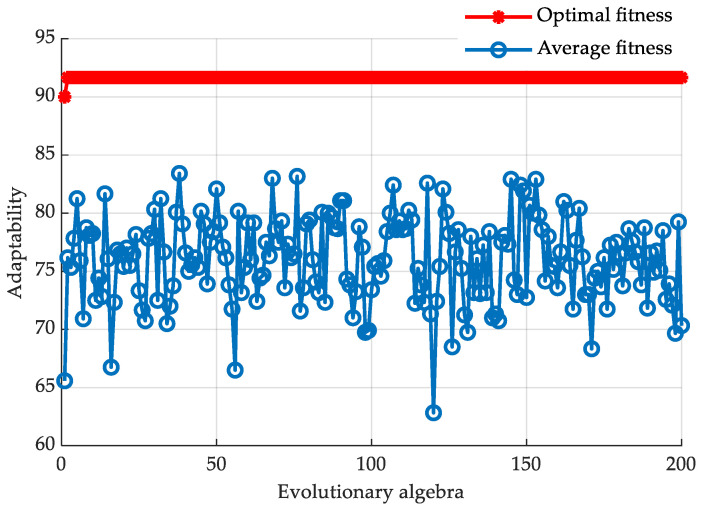
Fitness curves of PSO classification.

**Figure 12 sensors-20-06533-f012:**
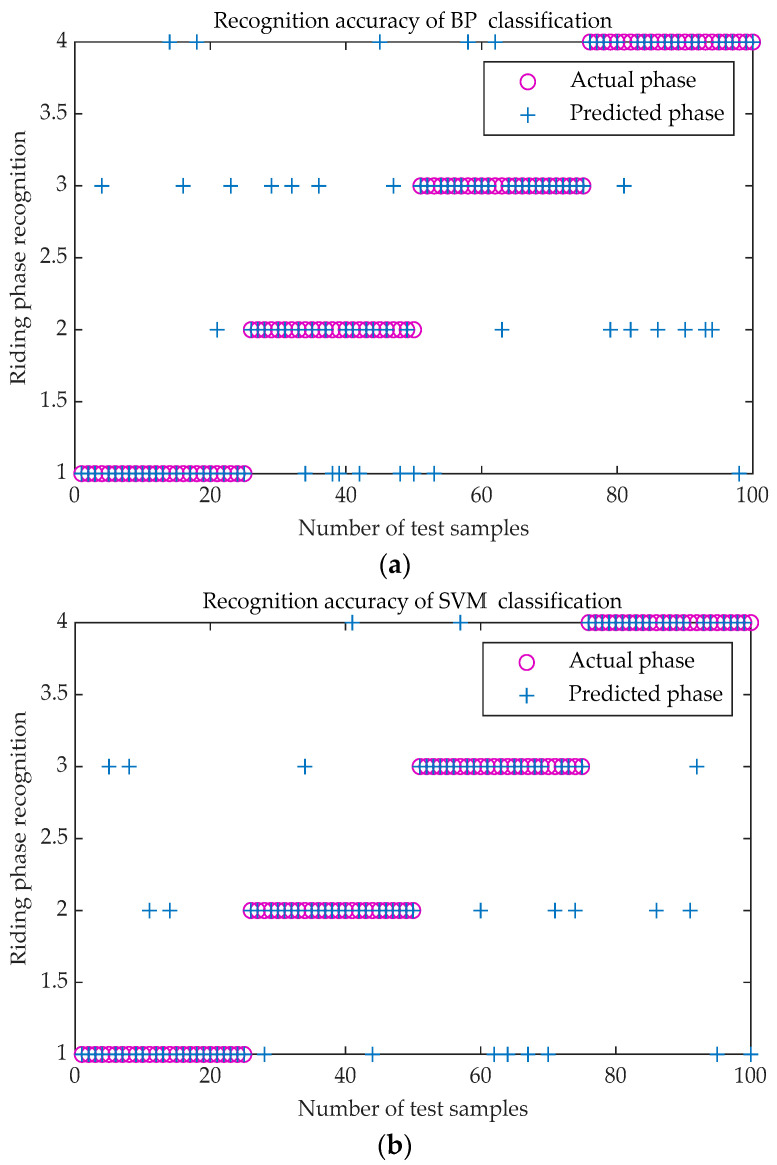
Comparison of recognition accuracy. (**a**) Recognition accuracy of BP classification; (**b**) Recognition accuracy of SVM classification; (**c**) Recognition accuracy of PSO-BP classification; (**d**) Recognition accuracy of PSO-SVM classification.

**Table 1 sensors-20-06533-t001:** Test results of different kernel functions.

Kernel Function	*C* = 2/*γ* = 1	*C* = 1/*γ* = 1	*C* = 2/*γ* = 0.5	*C* = 3/*γ* = 2
Polynomial s = 0	75%	73%	48%	83%
RBF s = 1	79%	85%	85%	83%
Linear s = 2	77%	83%	80%	85%
Sigmoid s = 3	31%	28%	82%	3%

**Table 2 sensors-20-06533-t002:** Cycling data recognition result.

Algorithm	Number of Samples					Recognition Rate (%)
	Pedaling	Lower Buffer	Relaxation	Upper Buffer	Sum	
BP	19	15	21	17	72	72.00
SVM	21	22	17	19	79	79.00
PSO-BP	20	20	21	23	84	84.00
PSO-SVM	23	24	23	23	93	93.00

**Table 3 sensors-20-06533-t003:** Precision.

Algorithm	Precision (%)			Average (%)	Standard Deviation (%)
	Classifier 1	Classifier 2	Classifier 3		
BP	72.00	70.67	76.00	72.89	2.26
SVM	79.00	77.33	72.00	76.11	2.98
PSO-BP	84.00	85.33	88.00	85.78	1.66
PSO-SVM	93.00	93.33	92.00	92.78	0.57

**Table 4 sensors-20-06533-t004:** Recall.

Algorithm	Recall (%)		
	Classifier 1	Classifier 2	Classifier 3
BP	76.00	60.00	84.00
SVM	84.00	88.00	68.00
PSO-BP	80.00	80.00	84.00
PSO-SVM	92.00	96.00	92.00

**Table 5 sensors-20-06533-t005:** F1-score.

Algorithm	F1-Score (%)		
	Classifier 1	Classifier 2	Classifier 3
BP	73.95	64.90	79.80
SVM	81.42	82.32	69.94
PSO-BP	81.95	82.58	85.95
PSO-SVM	92.5	94.65	92.00

**Table 6 sensors-20-06533-t006:** G index.

Algorithm	G Index (%)		
	Classifier 1	Classifier 2	Classifier 3
BP	73.97	65.12	79.90
SVM	81.46	82.49	69.97
PSO-BP	81.96	82.62	85.98
PSO-SVM	92.50	94.66	92.00

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
