# Peer review of "Bicycling Phase Recognition for Lower Limb Amputees Using Support Vector Machine Optimized by Particle Swarm Optimization"

_sensors, 2020, doi:10.3390/s20226533_

Round 1
Reviewer 1 Report
This paper proposes a novel method for recognizing the phases in bicycling of lower limb amputees using support vector machine (SVM) optimized by particle swarm optimization (PSO) is proposed in this paper. The method is essential for enhanced prosthetic knee joint control for lower limb amputees in carrying out bicycling activity. The recognition accuracy of the PSO-SVM classification model on testing data is 93%, which is much higher than those of BP, SVM and PSO-BP classification models. The particle swarm optimization (PSO) algorithm proposed in this paper is innovative in the recognition of cycling stage, but the following problems should be paid attention to:
- The author should make more elaboration on the contrast between the innovation points and traditional methods in order to highlight the advantages and innovation of the methods adopted in this paper.
- The angular velocity signals of prosthetic joints should be taken into account in experimental design and data collection to make the experimental data more convincing and reliable.
- The principle and method of noise filtering should be described more, so that the reader can understand more easily.
- In Section 3.3 of the paper, the author ignores the influence of bicycle seat height on knee joint angle, which may affect the size of joint angle and cannot be ignored.
- Graphics and tables can be colored differently to make the charts clearer and more beautiful in articles.
- The author should pay special attention to English grammar, spelling and sentence structure, and try to make the sentences simple and easy to understand, so that the reader can clearly understand the objectives and results of the research.
Author Response
Reviewer#1, Concern # 1: The author should make more elaboration on the contrast between the innovation points and traditional methods in order to highlight the advantages and innovation of the methods adopted in this paper.
Author response: We really appreciate your efforts and comments on our manuscript. We have revised our manuscript according to your comments and suggestions. The PSO-SVM method is mainly used in the article, other methods are only comparative experiments, so there is not much description of the traditional methods in the first draft.
Author action: We updated the manuscript by adding the following parts with yellow high lighting indicating changes.
(1) In Section 5, Paragraph 3: …BP neural network is trained and tested by data feature sets of known categories [29]. The BP neural network first sends the data to the input layer, and then passes it to the output layer through the hidden layer. Finally, the corresponding value and the global error between the value and the expected value are obtained in the output layer. In the continuous iteration process, the global error value is compared with the expected value. The set expected values are compared, and when the set conditions are met, the iteration is stopped, the connection weights and thresholds of each node in the BP neural network are output to establish a neural network classification model.
(2) In Section 5, Paragraph 6: Although the parameters of BP neural network can be optimized by PSO algorithm, the structure of BP neural network is based on empirical risk minimization, and its network connection weights are easy to fall into local minimization, which makes the training process appear under learning or over learning state with the change of prediction error. SVM is based on the principle of structural risk minimization. Compared with the traditional BP neural network, SVM has higher generalization performance and stability. The above results show that the method of using PSO optimized SVM method to realize riding phase recognition is effective, and has high recognition rate and practical reference value. …
([29] Li Z, Fan B H, Wang X, et al. Biomimetic Eye Positioning and Tracking Algorithm Based on BP Neural Network [J]. Robot, 2017, 39 (01): 63-69.)
Reviewer#1, Concern # 2: The angular velocity signals of prosthetic joints should be taken into account in experimental design and data collection to make the experimental data more convincing and reliable.
Author response: Thanks for your suggestion. During the experiment, we tried to obtain the angular velocity signal, but the angular velocity signal obtained by differentiating the angle of the prosthetic joint has irregular vibrations at the moment of phase transition, which cannot be used, and there is no wearable angular velocity sensor in the experiment.
Author action: Because the angle changes unevenly, the signal will jitter when the difference is calculated, which cannot be used, so the angular velocity signal is not added.
Reviewer#1, Concern # 3: The principle and method of noise filtering should be described more, so that the reader can understand more easily.
Author response: Thanks for your kind comments. We have supplemented the description of noise filtering.
Author action: We updated the manuscript by adding the following parts with yellow high lighting indicating changes.
In Section 3.2, Paragraph 2: …The improved wavelet packet denoising method is adopted to process the cycling information. This method can be summarized as combining the original single hard threshold or soft threshold denoising method. Firstly, the variance of noise signal and the wavelet transform coefficient are used to determine the threshold value. Secondly, the soft-hard threshold denoising function is used to determine the threshold value of wavelet packet coefficient. And the optimal solution of wavelet packet coefficient is obtained. Finally, the denoised signal is constructed by inverse transform with the decomposition coefficient of low-level wavelet packet, …
Reviewer#1, Concern # 4: In Section 3.3 of the paper, the author ignores the influence of bicycle seat height on knee joint angle, which may affect the size of joint angle and cannot be ignored.
Author response: Thanks for your valuable suggestion. For different prosthetic wearers, we will adjust the height of the seat before riding to try to make the knee angle of different participants have a small gap in each phase.(
Author action: We updated the manuscript by adding the following parts with yellow high lighting indicating changes.
In Section 3.3, Paragraph 1: …Compared with different prosthetic wearers, the seat height will be adjusted before riding. After the riding survey of several knee prosthetic wearers participating in the training, it is found that although the final seat height is different due to factors such as height, the angle difference of the knee joint in each phase is very small. …
Reviewer#1, Concern # 5: Graphics and tables can be colored differently to make the charts clearer and more beautiful in articles.
Author response: Thanks for your kind suggestion. Color pictures do make the article look better. We have made some changes to the form of some pictures.
Author action: We did color processing on the Yellow highlighted part of the picture, including:
(1) In Section 3.2:
Figure 4. Knee joint horizontal acceleration processing.
Figure 5. Knee joint vertical acceleration processing.
Figure 6. Ankle joint horizontal acceleration processing.
Figure 7. Ankle joint vertical acceleration processing.
Figure 8. Knee angle signal.
(2) In Section 4.3:
Figure 11. Fitness curves of PSO.
(2) In Section 5:
Figure 12. Recognition accuracy.
Reviewer#1, Concern # 6: The author should pay special attention to English grammar, spelling and sentence structure, and try to make the sentences simple and easy to understand, so that the reader can clearly understand the objectives and results of the research.
Author response: Thanks for your kind suggestion. We checked the grammar, spelling and sentence structure of the article, and made appropriate modifications.
Author action: We modify the sentences in the yellow highlighted part of the article.
(1) In Section 1, Paragraph 3: …A kind of prosthetic exoskeleton for riding has designed by Chaichaowarat et al. at Tohoku University in Japan. This prosthesis can store the energy of knee joint contraction in a torsion spring. Then the energy was released when the knee joint is extended. Thus, the fatigue of the leg muscles during repeated contraction exercises is reduced. …
(2) In Section 1, Paragraph 3: …Maaref et al. at University of Alberta in Canada presented a bicycle-type rehabilitation training robot. A method based on Gaussian mixture model is used in this robot. This method is also suitable for the study of prosthetic limb, and has a good reference value…
(3) In Section 1, Paragraph 3: … Actually, the prosthetic knee joint should adopt different control strategies according to the different dynamics features in each bicycle phase. Then, its energy-saving performance and control effect will be improved.
(4) In Section 1, Paragraph 4: In order to improve the accuracy of bicycle riding phase recognition, a multi-sensor system is proposed in this paper. The horizontal and vertical acceleration signals of knee joint and ankle joint as well as knee joint angle signal are collected to form multi-dimensional feature vector.
(5) In Section 1, Paragraph 4: The prosthetic knee joint can adjust its driving force according to the identified riding phase. This can make the riding action more fluent, higher energy utilization, and save the amputee's physical strength.

Reviewer 2 Report
Authors proposed a novel method to identify the phases during bicycling in lower limb amputees. Nine people were enrolled in the study and comparison among four different algorithms has been carried out, demonstrating that the SVM optimized by PSO is the best performing. Topic is interesting and in line with the Journal aim. However, the manuscript did not follow a precise scientific methodology.
Authors should consider the following issues to be solved:
- Authors should stress why the identification of all the bicycling phases is important;
- Authors should provide further details on the protocol:
- Did the average speed set by the operator or choose by the participant?
- Which kind of amputees are considered?
- Was the prosthesis the same for all the participants? Otherwise can different prostheses produce a bias in the results?
- Please provide the compliance of the protocol with the Helsinki Declaration
- The computation of the knee joint angle by means of inertial units require the application of a biomechanical model. Authors should better explain the procedure for the computation or better explain which angle was considered (the absolute one of the sensor?)
- Authors should provide information related to the kernel function for the SVM: linear, cubic?
- Authors should underline if each split of the classification was performed with the same SVM (same kernel, same iteration etc).
- It is not clear why only the SVM was described in detail. Even though the SVM has be found as the best performing, I strongly believe that also the NN has to be described in the methods.
- The applied validation is not robust. In fact, it is demonstrated that the cross-validation is the best way for testing the performance of a machine learning algorithm. I strongly suggest to split the dataset in two (training and test), as already done by the authors, but also perform the validation in turn in order to have all data as test at least one time.
- By applying cross-validation as suggested in point7, authors can obtain for each classifier the mean and standard deviation of the accuracy and perform a statistical analysis to understand if the best performance of the SVM are statistically significant and not casual due to the division between training and test dataset.
- Authors should consider also other parameters for the analysis of classifier performance: for example precision, F1-score, ROC curve, G index. To only compute the recognition rate (well-known as accuracy) did not allow to have a full validation of the classifier.
Author Response
Reviewer#2, Concern # 1: Authors should stress why the identification of all the bicycling phases is important.
Author response: We really appreciate your efforts and comments on our manuscript. We have revised our manuscript according to your comments and suggestions. Phase recognition is important for controlling, saving energy, and reducing the fatigue of amputees. We added a description of the importance of phase recognition.
Author action: We updated the manuscript by adding the following with yellow highlighting indicating changes.
In Section 1, Paragraph 1: …According to the recognized phase, the prosthetic joint could be control in an efficient way. In different stages of riding, the above-knee amputee's joints are in different states, the motor output power is also different, and the knee joint angle will change. All control strategies are based on the correct recognition of bicycle riding phase. …
Reviewer#2, Concern # 2: Authors should provide further details on the protocol:
Did the average speed set by the operator or choose by the participant?
Which kind of amputees are considered?
Was the prosthesis the same for all the participants? Otherwise can different prostheses produce a bias in the results?
Author response: Thanks for your valuable suggestion. The average speed of the participants in the article is selected by their own habits; the above-knee amputees are considered; all participants have the same prosthetic brand. In theory, our method is also applicable to other brands of prosthetics; The journal submitted proof materials in compliance with the declaration.
Author action: We updated the manuscript by adding the following parts with yellow high lighting indicating changes.
In Section 3.2, Paragraph 1: The 5 above-knee amputee volunteers in the experiment are 165-175cm high and 25-40 years old, 3 males and 2 females. They all have worn the Ottobock prosthesis for more than 5 years. In order to protect the safety of volunteer, a bicycle was placed on a shelf for data collection. The volunteers are asked to ride at an average speed for five minutes. The average speed is chosen by its own habits. As shown in Figure 3.
Reviewer#2, Concern # 3: The computation of the knee joint angle by means of inertial units require the application of a biomechanical model. Authors should better explain the procedure for the computation or better explain which angle was considered (the absolute one of the sensor?)
Author response: Thanks for your kind suggestion. For the calculation of the knee joint angle, we use the MPU9250 gyroscope wireless attitude sensor produced by Witt Intelligent Company. This sensor can calculate the collected signal in its internal, and then transmit it to the host computer, without further calculations. Therefore, the calculation process is not explained in the article.
Author action: We have supplemented the description of knee joint angle measurement in the yellow highlighted part.
In Section 3.1, Paragraph 3: …The calculation of the MPU9250 sensor is based on nine-axis signals. This sensor including three acceleration signals, three angular velocity signals, and three geomagnetic signals. It can calculate the collected knee joint angle data internally. Then, these data are transmitted into the host computer without further calculations.
Reviewer#2, Concern # 4: Authors should provide information related to the kernel function for the SVM: linear, cubic?
Author response: Thanks for your kind comments. The choice of the SVM kernel function is very important for the identification of the riding phase. We chose the RBF kernel function. We have supplemented the information of the kernel function.
Author action: We updated the manuscript by adding the following parts with yellow high lighting indicating changes.
In Section 4.1, Paragraph 2: SVM completes complex operations in low-dimensional space through nonlinear transformation, and establishes an optimal classification surface in high-dimensional space, and this nonlinear change is completed by the kernel function. Therefore, the selection of the SVM kernel function is very important. Through multiple comparison tests of different kernel functions, the RBF kernel function have a high recognition accuracy rate. Therefore, the Gaussian kernel function (RBF) is selected in this paper.
Reviewer#2, Concern # 5: Authors should underline if each split of the classification was performed with the same SVM (same kernel, same iteration etc).
Author response: Thanks for your kind suggestion. In the experiment, we used the same SVM (same kernel, same iteration etc), which we emphasized in the article.
Author action: We updated the manuscript by adding the following parts with yellow high lighting indicating changes.
In Section4.2, Paragraph2:The three SVM classifiers are the same.
Reviewer#2, Concern # 6: It is not clear why only the SVM was described in detail. Even though the SVM has be found as the best performing, I strongly believe that also the NN has to be described in the methods.
Author response: Thanks for your suggestion. Since the PSO-SVM method is mainly used in the article, there is not much description about NN in the article. We have supplemented the related content of neural network.
Author action: We updated the manuscript by adding the following parts with yellow high lighting indicating changes.
In Section 5, Paragraph 3: …BP neural network is trained and tested by data feature sets of known categories [29]. The BP neural network first sends the data to the input layer, and then passes it to the output layer through the hidden layer. Finally, the corresponding value and the global error between the value and the expected value are obtained in the output layer. In the continuous iteration process, the global error value is compared with the expected value. The set expected values are compared, and when the set conditions are met, the iteration is stopped, and the connection weights and thresholds of each node in the BP neural network are output to establish a neural network classification model. …Five input layer nodes represent five signal feature values, and four output nodes correspond to four phases. …
([29] Li Z, Fan B H, Wang X, et al. Biomimetic Eye Positioning and Tracking Algorithm Based on BP Neural Network [J]. Robot, 2017, 39 (01): 63-69.)
Reviewer#2, Concern # 7: The applied validation is not robust. In fact, it is demonstrated that the cross-validation is the best way for testing the performance of a machine learning algorithm. I strongly suggest to split the dataset in two (training and test), as already done by the authors, but also perform the validation in turn in order to have all data as test at least one time.
Author response: Thanks for your kind suggestion. Although it was not explained in the first draft, we conducted cross-validation during the process of splitting the data into two for training and testing.
Author action: We updated the manuscript by adding the following parts with yellow high lighting indicating changes.
In Section 5, Paragraph 2: Cross-validation was used in this experiment.
Reviewer#2, Concern # 8: By applying cross-validation as suggested in point7, authors can obtain for each classifier the mean and standard deviation of the accuracy and perform a statistical analysis to understand if the best performance of the SVM are statistically significant and not casual due to the division between training and test dataset.
Author response: Thanks for your valuable suggestion. Following the previous point, we have calculated the average and standard deviation of the accuracy of each classification method, and displayed them in the form of a table. The final result shows that PSO-SVM has the best performance.
Author action: We updated the manuscript by adding the following parts with yellow high lighting indicating changes.
Please see the Table 2 in Section 5.
Reviewer#2, Concern # 9: Authors should consider also other parameters for the analysis of classifier performance: for example precision, F1-score, ROC curve, G index. To only compute the recognition rate (well-known as accuracy) did not allow to have a full validation of the classifier.
Author response: Thanks for your kind comments. We have added F1 score and G index to the article to make the article more convincing.
Author action: We updated the manuscript by adding the following parts with yellow high lighting indicating changes.
(1) In Section 5, Paragraph 5: … The F1-score obtained by each SVM classification is shown in Table 4, and the G index is shown in Table 5. …
2) In Section 5, the following tables are presented.
Table 2. Precision.
Table 3. Recall.
Table 4. F1-score.
Table 5. G index.
(3) In Section 5,Paragraph 7:The PSO-SVM classification model has high F1-score and G index, and good recognition rate. It has important practical significance for the control of lower limb prosthesis in bicycle riding. …

Round 2
Reviewer 1 Report
The revised paper was improved well.
The contents of this paper might be more suitable for Article, not Letter.
Author Response
Reviewer#1, Concern # 1: The contents of this paper might be more suitable for Article, not Letter.
Author response: We really appreciate your efforts and comments on our manuscript again. We agree with your suggestion.
Author action: We will send a e-mail to the editor to change the paper type.

Reviewer 2 Report
Authors have properly answered to the previous highlighted issues.
I have only one minor comment before the acceptance. Authors should better describe the parameters used for assessing the performance of the classifier (meaning, equation, reference, range of variation ..). In addition, Table 2 and Table 3 are not cited in the main text.
Author Response
Reviewer#2, Concern # 1: I have only one minor comment before the acceptance. Authors should better describe the parameters used for assessing the performance of the classifier (meaning, equation, reference, range of variation ..). In addition, Table 2 and Table 3 are not cited in the main text.
Author response: We really appreciate your efforts and comments on our manuscript again. We have revised our manuscript according to your comments. We have provided a supplementary explanation on the parameters of the classifier in the article, and quoted Table 2-6.
Author action: We updated the manuscript by adding the following parts with yellow high lighting indicating changes.
(1)In Section 4.1, Paragraph 3: The kernel bandwidth γ is an important parameter of the kernel function, which affects the complexity and classification accuracy of the SVM model. The penalty factor C has the effect of adjusting the complexity of the model and can effectively improve the generalization performance of the model. The changes of the penalty factor C and the kernel bandwidth parameters γ can affect the classification accuracy of the SVM model. Therefore, different C and γ are used for comparative testing. The final result shows that the changes in the penalty factor and kernel function bandwidth have a significant impact on the model classification accuracy. The recognition rates for different kernel functions, different kernel bandwidth and penalty factors are shown in Table 1.
Table 1. Test results of different kernel functions.
After comparing the experimental results, it can be seen that the Gaussian Kernel Function (RBF) has a high recognition rate during the recognition process, which can be maintained above 79%, and is relatively stable. Therefore, the Gaussian kernel function (RBF) is selected in this paper. …
(2) In Section 5, Paragraph 5: …The precision of each SVM classification is shown in Table 3. The recall of each SVM classification is shown in Table 4. …
[As a new Table is added to describe the parameters of SVM, so the original Table No.2 and 3 increase as No. 3 and 4. ]
(3) In Section 5, Paragraph 7: As shown in Table 2, PSO-SVM has high recognition rate; As shown in Table 3, PSO-SVM has the highest precision and the smallest standard deviation among the four classification models; As shown in Table 4, PSO-SVM has high recall; As shown in Table 5-6, the PSO-SVM classification model has high F1-score and G index. The above results show that the method of using PSO optimized SVM method to realize riding phase recognition is effective, and practical reference value.
